# A Comparative Analysis of Emotional Regulation and Maladaptive Symptoms in Adolescents: Insights from Iran and Belgium

**DOI:** 10.3390/healthcare12030341

**Published:** 2024-01-30

**Authors:** Shokoufeh Vatandoost, Imke Baetens, Zeinab Erjaee, Zahra Azadfar, Martijn Van Heel, Lisa Van Hove

**Affiliations:** 1Department of Psychology, Faculty of Psychology & Educational Sciences, Vrije Universiteit Brussel, 1050 Brussels, Belgium; lisa.van.hove@vub.be; 2Brussels University Consultation Center, Department of Psychology, Faculty of Psychology & Educational Sciences, Vrije Universiteit Brussel, 1050 Brussels, Belgium; imke.baetens@vub.ac.be (I.B.); zahra.azadfar@vub.be (Z.A.); martijn.van.heel@vub.be (M.V.H.); 3Department of General Psychology, Faculty of Humanities and Social Sciences, Science and Research Branch, Islamic Azad University, Tehran 37541-374, Iran

**Keywords:** emotional regulation, internalizing symptoms, externalizing symptoms, NSSI, adolescent, culture

## Abstract

(1) Background: Emotional regulation is a critical determinant of adaptive functioning during adolescence, exerting a profound influence on psychological well-being. This study seeks to deepen our understanding of the intricate interplay between emotional regulation and maladaptive psychological symptoms, examining these dynamics through a cross-country comparison. (2) Methods: A total of 224 adolescents, aged 13 to 21 years, from both Iran and Belgium, participated in a cross-sectional comparative study. The study aimed to elucidate the relationship between emotional regulation and mental health functioning, assessing internalizing symptoms, externalizing symptoms, and self-injury. A demographic questionnaire, the Emotion Regulation Inventory, the Strengths and Difficulties Questionnaire, and the Self-Harm Inventory, were administered. Data analysis incorporated correlation assessments, multivariate analysis of variance, and structured equation modeling. (3) Results: The findings revealed a positive association between emotional dysregulation and psychological symptoms across the entire sample. Conversely, emotional suppression, more prevalent in Iran, showed no significant link with maladaptive symptoms but was associated with self-harm in the Belgian sample. Cultural disparities were evident, with internalizing problems more prevalent in Iran and externalizing issues more common in Belgium. (4) Conclusions: Emotional dysregulation emerged as a common factor compromising mental health. It emphasizes the necessity of considering cultural nuances when developing interventional and preventative programs and calls for further research in this field.

## 1. Introduction

Emotional regulation (ER)—encompassing the internal or external process of monitoring, evaluating, and modulating emotional experiences [1,2]—constitutes a crucial psychological resource with a significant impact on mental health and social adjustment [3,4,5]. Although the importance of ER is widely recognized [6,7,8,9], there exists limited consensus on the different dimensions of ER and their distinct associations with psychopathology [10]. 

Based on the self-determination theory (SDT), Roth et al. [11] introduced the concept of Integrative Emotion Regulation (IER). SDT is a comprehensive motivational theory aimed at elucidating the relationship between psychological development and social adaptation [12]. Within the SDT taxonomy of ER styles, IER is characterized by an inclination toward openness and receptivity to one’s own emotional states. This approach involves the impartial processing of emotions and the deliberate cultivation of full emotional awareness. In contrast, suppressive emotion regulation (SER) represents a controlled form of ER that hinges on the avoidance of emotional experiences with the goal of mitigating the impact of emotional inputs. The third mode of ER is referred to as dysregulated emotion regulation (DER), wherein emotions are perceived as overwhelming, and the individual grapples with difficulties in effectively managing these emotional states [13]. These distinctions within ER styles, particularly the introduction of IER, contribute significantly to our understanding of how individuals navigate their emotional landscapes and how these approaches may impact mental health and social adaptation. 

ER has been considered as playing a particularly salient role in the context of adolescent mental health [14], a developmental phase distinguished by heightened emotional intensity and heightened moment-to-moment variability in emotional experiences [15,16]. The research has consistently documented the detrimental effect of maladaptive ER strategies (i.e., SER and DER) on adolescents’ mental health [17,18,19]. In contrast, IER has been shown to decrease the risk of developing psychopathology in adolescents [20,21,22]. This emerging body of research highlights the importance of further exploring the distinctive role of ER styles in adolescent psychopathology.

Among the most prevalent mental health problems among adolescents is non-suicidal self-injury (NSSI), which has been shown to be strongly related to difficulties in ER [23,24,25]. NSSI is defined as “the deliberate, non-fatal self-infliction of body tissue damage without suicidal intent” [26], and is associated with adverse outcomes, including diminished academic performance, peer-related difficulties, and an elevated risk of suicidal behaviors [27,28,29]. 

Theoretical frameworks propose that challenges in ER are also pivotal in the development of internalizing symptoms among adolescents [30,31,32]. Building on this notion, the research consistently indicates that difficulties in ER predict the emergence of both internalizing and externalizing symptoms in adolescents [33,34,35]. Moreover, extending our understanding of adolescent mental health, previous studies have established a clear link between NSSI and internalizing and externalizing symptoms [36,37]. As proposed in the Four-function Model by Nock and Prinstein [38], NSSI is considered as a maladaptive means of coping with negative emotions and thoughts. Difficulty tolerating intense emotions can lead to a tendency to internalize emotional distress, which in turn increases susceptibility to non-suicidal self-injury (NSSI). Delving deeper into the dynamics of ER and its impact, it has been observed that the frequency of NSSI is closely associated with emotional dysregulation. Importantly, this relationship is assumed to be mediated by the presence of internalizing symptoms [39]. 

However, the role of internalizing and externalizing symptoms as underlying mechanisms in the relationship between integrative, suppressive, and dysregulated ER with NSSI has not yet been examined. Furthermore, the vast majority of studies in this area are limited to Western adolescents. 

Previous studies, distinct from the novel SDT taxonomy, have elucidated cultural disparities in ER and emotional expression [40,41]. Nevertheless, it is essential to highlight that, so far, only Benita et al.’s study [20] has delved into the SDT taxonomy of ER, encompassing the examination of integrative, suppressive, and dysregulated ER as universal constructs. As emphasized in a meta-analysis review by Hu et al. [42], examining the relationship between ER and mental health, it is very important to consider culture as a moderator in this regard. It is noteworthy that Western cultures, in general, tend to exhibit a leaning toward individualism when compared to Eastern cultures, with Iran occupying an intermediary position between the individualistic and collectivist cultural spectrums [43,44]. Iranian families exhibit a profound cultural inclination toward the socialization of their offspring, instilling a deeply ingrained reverence for elder family members. Moreover, they emphasize the preservation of familial harmony by nurturing a strong sense of filial respect toward parents and other kin. Within this cultural context, children are subtly discouraged from openly expressing dissenting opinions or disagreements within the family dynamic [45,46]. Belgium is grouped as a Western individualist culture [47] and individualism predicts more emotional expression and personal independence in the socialization of emotions [41,48]. 

### The Present Study 

The present study aims to investigate the cross-cultural relevance of ER styles in the context of adolescent psychopathology. To achieve this goal, we have formulated three main objectives: 

**Objective** **1:**
*We examined the relationship between three different ER styles (IER, SER, and DER) and adolescent psychopathology (internalizing and externalizing problems and NSSI), as well as cultural differences in these relationships.*


**Hypothesis** **1a.**
*We expect a significant negative relationship between IER and psychological symptoms and a significant positive relationship between DER and SER and psychological symptoms in the two cultural groups.*


**Hypothesis** **1b.**
*We expect to find a significant correlation between higher levels of suppression and increased psychological symptoms. Furthermore, we anticipate that this correlation will be more pronounced and negatively impactful in the Belgian context.*


**Objective** **2:**
*We examined the mediating role of internalizing and externalizing problems in the association between ER styles and NSSI.*


**Hypothesis** **2.**
*We hypothesized that internalizing and externalizing problems would mediate the relationship between ER styles and NSSI in both cultural groups where SER and DER heighten the probability of NSSI through the escalation of the internalizing and externalizing of symptoms and IER is expected to reduce the likelihood of NSSI by diminishing the internalizing and externalizing of symptoms.*


**Objective** **3:**
*We examined the role of ER styles in predicting the internalizing and externalizing of problems and NSSI, controlling for age, gender, and country.*


**Hypothesis** **3.**
*We hypothesized that difficulties with emotional regulation (i.e., SER and DER) would predict psychological outcomes (NSSI, internalizing and externalizing symptoms) above and beyond age, gender, and country.*


## 2. Materials and Methods

### 2.1. Participants

This study included a total of 224 participants from secondary school students in Iran (Tehran, n = 117, 38% (44 of 117) female) and Belgium (Antwerp, n = 107, 52% (56 of 107) female). The age range in the Iranian sample spanned from 13 to 15 years, with a mean age of 13.85 (SD = 0.62); meanwhile, in the Belgian sample, the age range was from 14 to 19 years, with a mean age of 15.23 (SD = 1.20). It is important to note that in Belgium, given the considerable ethnic diversity, only adolescents with both parents born in Belgium were included in the study. They made up 30.83% (n = 107) of the total sample of 347 Belgian participants. 

### 2.2. Measures

Demographic questions included age (13–19 years), sex (male and female), and country of origin, including parents’ place of birth. 

The self-report version of the Strengths and Difficulties Questionnaire [49] for ages 11–17 years was used to measure internalizing and externalizing symptoms. Goodman and Goodman [50] recommended the use of three subscales in non-clinical samples: *Internalizing Problems* (emotional symptoms (e.g., “*I worry a lot*”) and peer problems (e.g., “*I am usually on my own*”), 10 items), *Externalizing Problems* (conduct problems (e.g., “*I fight a lot*”) and hyperactivity symptoms (e.g., “*I am constantly fidgeting*”), 10 items), and the *Prosocial Subscale* (e.g., “*I try to be nice to other people*”; 5 items). Each item is rated on a 3-point Likert scale, where 0 = *not true*, 1 = *somewhat true*, and 2 = *definitely true*. The total difficulties score is computed by adding up scores from all scales except the prosocial scale. The resulting score falls within the range of 0 to 40. A score between 16 and 19 is categorized as borderline and between 20 and 40 is in the clinical range. Only the total score and the subscales of internalizing and externalizing problems were used in this study. Previous studies have reported satisfactory psychometric properties for the Dutch [51] and the Persian versions [52] of the self-report version of the SDQ. In the study by Goodman [53], the internal consistency for the total score was 0.73 and good convergent validity (*r* = 0.70 for the total score) was reported. In the current study, the Cronbach’s alpha for the total score, internalizing and externalizing problems, was 0.68, 0.71, and 0.72, respectively.

Emotion Regulation Inventory [54]. This questionnaire includes three styles of ER: (1) *dysregulated* (6 items, e.g., “*When I am angry, I feel I have little control over my behaviour*”), (2) *suppressive* (6 items, e.g., “*I almost always try to not express my anger* ”), and (3) *integrative* (6 items, e.g., “*When I am angry, I usually try to understand why I am angry*”). These items are rated on a 5-point Likert scale ranging from 1 (*not at all*) to 5 (*very much*). Roth et al. [54] found evidence for the validity and internal consistency of all three subscales. Using the Dutch version of the ERI, Brenning et al. [55] found Cronbach’s alphas of 0.69 for dysregulated and 0.72 for suppressive ER. The Cronbach’s alphas in the current study were 0.76, 0.73, and 0.85 for DER, SER, and IER, respectively. For the Iranian sample, the English version of the ERI [54] was translated into Persian by a Persian-speaking researcher and then back-translated by another researcher. In line with the suggestions made by Sousa and Rojjanasrirat [56], we implemented a systematic approach that involved several key steps: 1. translating the original instrument into the target language (forward translation or one-way translation), 2. comparing the two translated versions of the instrument, 3. conducting a blind back-translation (blind backward translation or blind double translation) of the preliminary initial translated version of the instrument, and 4. comparing the two back-translated versions of the instrument to ensure comparability between the starting and back-translated versions. The Cronbach’s alpha for this Persian version was 0.63, 0.80, and 0.67 for the DER, SER, and IER subscales, respectively. 

Self-Harm Inventory [57]. This measure assesses the respondent’s history of indirect self-injurious behaviors, NSSI, and suicidal self-injurious (SSI) behaviors with a 23-item (yes/no) questionnaire. Each item is preceded by a question: “*Have you ever intentionally, or on purpose*…”. In this study, we used only the 8 items examining NSSI methods (including methods such as cutting, banging, scratching, and burning) and a single SSI item (i.e., “*Have you ever intentionally, or on purpose, attempted suicide?*”). Baetens et al. [58] documented an internal consistency of 0.63 for the Belgian sample, and Tahbaz et al. [59] reported a Cronbach’s Alpha of 0.74 for the Iranian sample. In the current study, the Cronbach’s alpha for the NSSI subscale was 0.69.

### 2.3. Procedure

This study was conducted using a convenience sample, where researchers approached a school with which they had personal connections. Out of the eight secondary schools contacted in the Antwerp region, three expressed interest in participating. All these schools, situated in a metropolitan setting, have an average enrollment of 488 students across three education levels. Out of the total student population of 488, data were successfully collected from 376 students, indicating a response rate of 77.04%. The primary focus of this study was to examine cultural distinctions between Iran and Belgium. To ensure a more homogeneous sample representative of the diverse population in Belgium, only participants with both parents born in Belgium were included.

Before conducting the tests, parents were sent passive informed consent via letter or email. Active informed consent was obtained from the students themselves at the start of the test administration. The informed consent emphasized the voluntary nature of participation, ensured the confidentiality of the data, and clarified that the students’ information would be coded for anonymity in subsequent years of the longitudinal study.

Participants were given 50 min to complete the questionnaires. Test administrators and research assistants circulated during completion of the questionnaire to match students with their corresponding sticker codes. Test leaders remained present throughout, addressing any questions. As a token of appreciation, participants received an apple and a lucky doll. Additionally, they received a letter containing more information about the research, and the contact details of the researchers. A parallel procedure was followed in Iran for the two schools in Tehran. The questionnaires were administered to the adolescents during their regular school hours under the supervision of a research assistant. It is important to note that the research protocol was approved by the Medical Ethics Committee of UZ Brussels (2019/073) and was also approved by Yazd Shahid Sadoughi University of Medical Sciences (IR.SSU.REC.1400.159) in Iran.

### 2.4. Statistical Analyses

The statistical analyses in this study were conducted using SPSS (version 25) and Amos software (version 24). In the first step, a thorough check for missing data was performed before conducting analyses, and one participant’s data were excluded due to an excessive number of unanswered questions. After these initial data checks, descriptive statistics were then calculated. The normality of the data was examined using the Shapiro–Wilks test. Subsequently, correlation analyses were conducted to examine the relationship between the three modes of ER, internalizing and externalizing problems, and NSSI. These correlations were calculated for the entire group of participants and separately for the Iranian and Belgian samples, allowing a comprehensive examination of the relationships between these variables within different cultural contexts. Cohen’s q effect sizes were calculated to examine the differences in correlations between two countries and were interpreted according to Cohen [60]: 0.10 = small effects, 0.30 = moderate effects, and 0.50 = large effects. 

Moving forward, a multivariate analysis of variance (MANOVA) was performed to explore potential differences between the Belgian and Iranian samples with regard to the mean scores of the ER modes and the internalizing and externalizing of problems. A hierarchical regression analysis was performed to assess the predictive relationship between emotional regulation (ER) modes and internalizing and externalizing problems, with control for demographic variables (age, gender, and country). Additionally, a hierarchical logistic regression analysis was conducted to investigate whether ER modes predict NSSI, accounting for these demographic variables. 

Finally, structural equation modelling (SEM) was employed to examine the direct effects between the variables and the indirect effects of ER modes on NSSI through the internalizing and externalizing of problems separately in the Belgian and Iranian samples. This advanced analytical method allows for a more in-depth examination of the complex interactions and pathways between these variables, facilitating a comprehensive understanding of their associations. 

Additionally, before initiating the study, we conducted a power analysis to determine the appropriate sample size. With an effect size of 0.30, alpha set at 0.05, and a desired power of 0.8, the calculated sample size was 50 for MANOVA and 53 for hierarchical regression analyses using GPower 3.1.9.7. Our actual sample size exceeded these requirements. Additionally, following the guidelines of Kellar and Kelvin [61], the recommended sample size for structural equation modeling (SEM) was 5–20 times greater than the total number of items in the measure. 

## 3. Results

### 3.1. Preliminary Analysis

Descriptive analyses (mean and standard deviations) and the correlation between ER modes, the internalizing and externalizing of problems, and NSSI are presented in Table 1. Pearson correlation results showed that both DER and SER were positively correlated with internalizing problems. In addition, DER was also positively correlated with externalizing problems and NSSI. SER, inversely, showed a negative correlation with externalizing problems and NSSI. However, IER showed no statistically significant correlations with the internalizing and externalizing of problems and NSSI. In terms of the correlation between internalizing and externalizing problems and NSSI, both internalizing and externalizing problems were positively associated with NSSI. It is worth noting that the most robust correlation is observed between externalizing problems and NSSI (r = 0.54, *p* < 0.01).

Table 2 presents the descriptive statistics and bivariate correlations between ER modes, the internalizing and externalizing of problems, and NSSI in the two different samples, each representing a different cultural context. In the Belgian sample, Pearson correlation results indicated that the maladaptive ER modes (SER and DER) were positively and significantly correlated with NSSI, while IER was negatively correlated with NSSI. Additionally, DER was positively correlated with both the internalizing and externalizing of problems. However, SER and IER were not significantly associated with the internalizing and externalizing of problems. In the Iranian sample, DER was positively significantly associated with all maladaptive symptoms (the internalizing and externalizing of problems and NSSI). SER was positively significantly associated with internalizing problems and unexpectedly negatively associated with externalizing problems. The correlation between SER and NSSI was not significant (r = −0.13, *p* > 0.05). IER showed no significant correlation with maladaptive symptoms (*p* > 0.05). Notably, internalizing and externalizing problems were strongly positively correlated with NSSI in both countries. 

The results of the Cohen’s q for the assessment of the differences in the correlations between two countries revealed small effect sizes for the correlation between externalizing problems with SER (r = 0.229) and IER (r = 0.256) and also the correlation between NSSI with DER (r = 0.114) and IER (r = 0.175). Medium effect sizes were observed for the association between SER and NSSI (r = 0.256) and also for the association between NSSI and externalizing problems (r = 0.256).

### 3.2. Multivariate Analysis of Variance

Before performing MANOVA, a series of tests including the Shapiro–Wilk test, Box’s M test, Levene’s test, and Wilks’ lambda were conducted to assess the assumptions. The results of the Shapiro–Wilk test indicated the normal distribution of all the variables in the data (*p* > 0.05). Following this observation, and the subsequent visual examination of the histogram of variables and the QQ plot, a decision was made to employ a parametric test. The Mbox test for analysis of variance–covariance matrices showed that the value of this test was not statistically significant (Box M = 21.836, F = 1.421, *p* = 0.127); therefore, the assumption of the homogeneity of variance–covariance matrices was observed. The results of the Levene’s test indicated no significant differences between the variances of the two groups in all the variables including DER (F = 0.022, *p* = 0.882), SER (F = 0.398, *p* = 0.529), IER (F = 1.896, *p* = 0.17), internalizing (F = 0.651, *p* = 0.421), and externalizing problems (F = 0.903, *p* = 0.343). In addition, the results of the Wilks’ lambda test showed that there was a significant difference in at least one of the variables between two groups (Wilks’ Lambda = 0.58, F = 31.627, *p* = 0.42).

The results of the MANOVA showed significant differences between Belgian and Iranian adolescents in terms of SER (F = 95.9, *p* < 0.01), IER (F = 15.04, *p* < 0.01), internalizing (F = 27.03, *p* < 0.01), and externalizing problems (F = 67.5, *p* < 0.01). More specifically, the Belgian adolescents scored significantly lower on SER and internalizing problems and higher on IER and externalizing problems compared to the Iranian adolescents (see Table 3). However, there were no significant differences in DER scores between the two countries (*p* > 0.05). A Chi-square goodness of fit test was employed to evaluate differences in non-suicidal self-injury (NSSI) between the two countries. The outcomes demonstrated a significant dissimilarity in NSSI proportions, X^2^(1, 224) = 45.15, *p* < 0.05, suggesting a higher likelihood of NSSI among Belgian adolescents compared to their counterparts in Iran.

### 3.3. Hierarchical Regression Analyses

The results of the hierarchical regression analysis showed that in the first step (Model 1), the combined impact of the control variables (i.e., age, gender, and country) accounted for a modest 4% of the variance in the internalizing and externalizing of problems and a substantial 16% of the variance in NSSI scores (see Table 4). However, when we introduced the ER modes in the next step (Model 2), there was a noticeable increase in the explained variance. Specifically, the cumulative variance explained for the internalizing and externalizing of problems increased to 27% and for NSSI to 30%. Both increments were statistically significant (*p* < 0.01). Among the ER modes, only DER emerged as a statistically significant predictor of both internalizing and externalizing problems (*β* = 0.49, *p* < 0.01) and NSSI (*β* = 0.35, *p* < 0.01) in Model 2. Among the control variables, the country had a significant effect on NSSI (*β* = 5.698, *p* < 0.01) and DER (β = 5.598, *p* < 0.01). None of the other variables examined played a significant role in predicting internalizing and externalizing problems and NSSI (see Table 4).

### 3.4. Mediation Analysis

The structural model with all significant paths for the mediating role of internalizing and externalizing problems in the association between ER modes and NSSI for the Belgian sample is shown in Figure 1. As the initially proposed model did not fit the data well, all insignificant paths were removed (i.e., the paths from SER and IER to externalizing problems and the path from DER to NSSI). Following the corrective indices suggested by the software, we established six covariance relationships among the errors of certain observed variables. Collectively, these adjustments brought the model fit to an acceptable level. The model fit indices for the presented model were satisfactory (χ^2^/df = 1.25, *p* = 0.02, CFI = 0.92, IFI = 0.93, RMSEA = 0.04). The values of the standardized regression weights for direct effects were statistically significant for all paths in the model, but for the path from SER to NSSI *(*β = 0.126, *p* < 0.05) the effect size was small. The results of the standardized indirect effects indicated a significant mediating role for internalizing and externalizing problems in the relationship between DER (β = 0.35, *p* > 0.05) and SER modes and NSSI (β = 0.07, *p* > 0.05). However, the association between IER and NSSI was only mediated by internalizing problems (β = 0.07, *p* > 0.05).

The structural model with all significant paths for the mediating role of internalizing and externalizing problems in the association between ER modes and NSSI for the Iranian sample is presented in Figure 2. As the initially proposed model did not fit the data well, all insignificant paths were removed (i.e., the path from DER to NSSI, the path from internalizing problems to NSSI, and IER to internalizing and externalizing problems and NSSI). The model fit indices for the presented model were satisfactory (*χ^2^*/df = 1.2, *p* = 0.06, CFI = 0.93, IFI = 0.94, RMSEA = 0.04). The values of the standardized regression weights for direct effects were statistically significant for all paths in the model; however, for the path from SER to NSSI (β = 0.18, *p* < 0.05) the effect size was small. The results of the standardized indirect effects revealed a significant mediating role of externalizing problems in the relationship between DER (β = 0.6, *p* > 0.05) and SER (β = −0.33, *p* > 0.05) modes and NSSI.

## 4. Discussion

In pursuit of our primary objective, an investigation into the interplay between emotional regulation (ER) styles, psychological symptoms, and cultural variations was conducted among adolescents sampled from both Iran and Belgium. As hypothesized (Hypothesis 1a), our findings uncovered positive correlations between emotional dysregulation and internalizing symptoms, externalizing symptoms, and non-suicidal self-injury (NSSI) in both cohorts, aligning with the existing literature [62,63,64] that consistently associates emotional dysregulation with adverse mental health outcomes across diverse cultures. Additionally, our results support the prevailing consensus that emotional dysregulation serves as a risk factor for adolescent psychopathology, as emphasized in the prior research, includingClapham et al. [65]. 

Contrary to our initial hypothesis, the relationship between the suppression and externalization of symptoms was significantly negative. This unexpected finding can be elucidated by studies [66,67] that have indicated that adolescents employing expressive suppression may decrease the sharing of emotions, experience a decline in social support, and diminish proximity to peers, thereby reducing the externalizing behavior typically observed in social interactions. Unexpectedly, no significant relationship between integrative regulation and psychological outcomes was identified, contradicting the findings of Benita [20] who posited that integrative regulation is a universal concept beneficial for mental health across various cultures.

Turning attention to the cross-cultural differences, our study, inspired by insights from Mesquita [68], highlighted the crucial role of cultural values, norms, and ideas in shaping psychological symptoms, emotional regulation, and normative behavior. Distinct patterns within the Iranian and Belgian samples emerged, with the Iranian cohort exhibiting higher occurrences of internalizing symptoms and a prevalent use of suppressive emotion regulation (SER) modes. This aligns with prior research by Dadkhah and Shirinbayan [69], indicating a substantial incidence of emotional suppression among Iranian adolescents, highlighting the robust influence of cultural norms on emotional regulation in collectivistic societies. 

Conversely, the Belgian sample displayed higher frequencies of externalizing symptoms and a greater inclination toward integrative emotion regulation (IER), reflecting the cultural emphasis on self-expression, self-reliance, and the pursuit of individual goals in Belgium’s individualistic context. Consistent with hypothesis 1b, Iranian participants showed significantly higher rates of emotional suppression, while the Belgian sample exhibited higher rates of integrative regulation styles, further emphasizing the impact of culture-specific norms on how to cope with emotions. 

Based on the sociodynamic model offered by Mesquita [68], emotional suppression as a strategy of regulation is not always dysfunctional, and may only be in cultural contexts that value authenticity (which is a characteristic of individualistic cultures). Indeed, our results show that emotional suppression was most prevalent in Iran, and showed no detrimental effects on psychological functioning, whereas emotional suppression in the Belgian sample was associated with a heightened risk for non-suicidal self-injury (NSSI); thus, echoing this model conceptualization and highlighting the importance of considering culture as an important factor in exploring the adaptiveness of emotional regulation strategies. Soto et al. [70] found that the negative associations between suppression and psychological functioning observed in European Americans were not necessarily seen among members of East Asian cultures, where emotional restraint is relatively encouraged over emotional expression. This suggests that the prevalence of emotional suppression in Iranian adolescents may be influenced by cultural norms that prioritize emotional restraint. 

These findings collectively suggest that the differences in emotional suppression and emotional regulation strategies between the Iranian and Belgian samples can be attributed to cultural variations in the acceptance and regulation of emotions.

In alignment with Hypothesis 2, mediation analysis revealed that internalizing and externalizing problems served as mediating factors in the association between DER and NSSI in the Belgian sample, with internalizing problems demonstrating a more pronounced influence. 

In the Iranian model, only externalizing problems acted as a mediator in the relationship between DER and NSSI. These outcomes align with prior research (for review, see [65]) demonstrating links between emotional dysregulation, psychological symptoms and problems, and NSSI. For example, the study by Baetens et al. [58] is pertinent, as it identified the significant association of externalizing problems with NSSI. This finding corroborates the mediating role of externalizing problems in the relationship between emotional dysregulation and NSSI, as indicated in Kranzler [39].

Supporting Hypothesis 3, the hierarchical regression results indicated that DER positively predicted NSSI and internalizing and externalizing problems beyond age, gender, and cultural differences. This underscores the robust correlation between DER and psychological symptoms in both countries, aligning with the conceptualization proposed by Clapham and Brausch [65] and recent studies (e.g., [71]) emphasizing emotional dysregulation as a key predictor of NSSI. In conclusion, our study contributes to the understanding of the association between emotional dysregulation and psychopathology across diverse countries, shedding light on the nuanced interplay of cultural factors in shaping emotional regulation styles and their impact on psychological well-being. This study exhibits several strengths, including its cross-cultural focus, theoretical framework, integration of previous research, comprehensive analysis, and relevance to cultural norms. These strengths collectively contribute to the study’s significance in advancing our understanding of the interplay between emotional regulation and psychopathology in diverse cultural contexts. However, there are also several limitations that should be considered in this study. First, our study only focusses on a Belgian and Iranian sample. Future cross-cultural studies should include several (and other) cultures to test the same research hypothesis, in order to increase the generalization of results and avoid extrapolation of the findings. 

Secondly, in this study, we tried to take into account processes of cultural adaption, and only selected adolescents with both parents born in Belgium, or Iran, respectively, and there was no measure of socio-economic status. Although we are well aware that this limits the generalizability of results in our multicultural societies, future research should broaden the scope and also examine potential differences in multi-cultural citizens. 

Furthermore, due to the cross-sectional design of this study, a causal relationship could not be assumed. Future cross-cultural studies should include longitudinal designs. 

Also, due to the high variability in psychological symptoms and well-being in adolescence, exploring the potential of ecological momentary assessment (EMA) technologies for the repeated real-time assessment of emotions represents an intriguing avenue for future research.

Additionally, the scope of our study is confined to a particular age group. Subsequent research endeavors stand to gain valuable insights by examining cultural variations across diverse age groups, thereby capturing potential age-related factors. 

Finally, the ERI has not been validated in Iran making it vulnerable to bias. Future validation studies in Iranian samples are needed.

## 5. Conclusions

Our study contributes to the growing body of research on ER modes and their relationship with adolescent psychopathology from an SDT perspective; examining the distinctive role of each ER modes in psychological symptoms of a sample of youth from two distinctive cultures. Our study underscores some cultural differences in the links between ER modes and psychological symptoms in adolescents. The most noteworthy cultural differences are the variations in levels of internalizing symptoms and emotional suppression between Iranian and Belgian adolescents. These results underscore the significance of considering cultural nuances in understanding the interplay between ER modes and adolescent psychopathology, further emphasizing the need for more extensive cross-cultural research to build a more comprehensive understanding of these phenomena.

## Figures and Tables

**Figure 1 healthcare-12-00341-f001:**
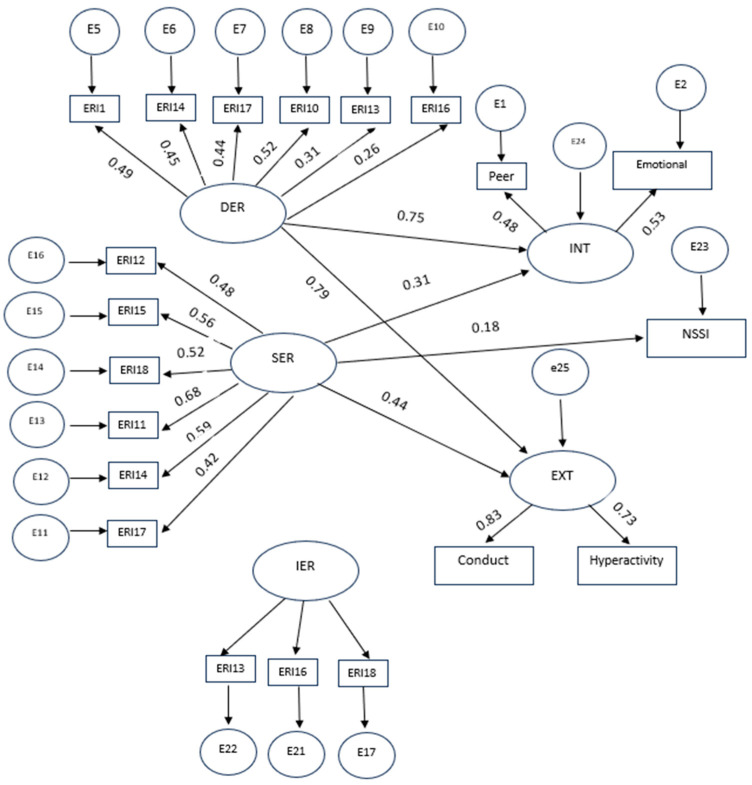
Structural model for the mediating role of internalizing and externalizing problems in the relationship between ER modes and NSSI in the Belgian sample. Note: DR stands for dysregulation; int stands for internalizing symptoms; SR stands for suppressive regulation; IR stands for integrative regulation; Ext stands for externalizing symptoms.

**Figure 2 healthcare-12-00341-f002:**
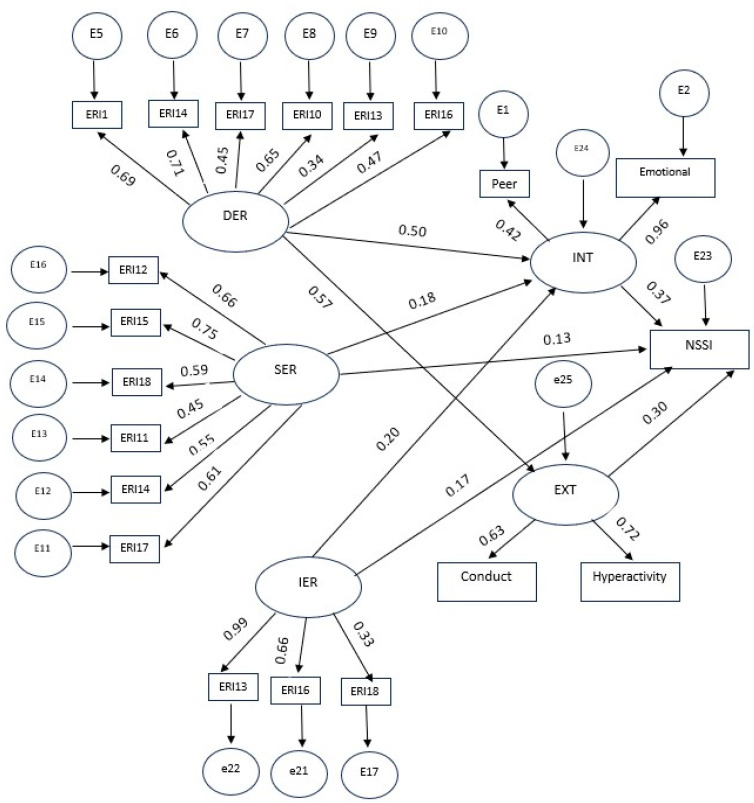
Structural model for the mediating role of internalizing and externalizing problems in the relationship between ER modes and NSSI in the Iranian sample. Note: DR stands for dysregulation; int stands for internalizing symptoms; SR stands for suppressive Regulation; IR stands for integrative regulation; Ext stands for externalizing symptoms.

**Table 1 healthcare-12-00341-t001:** Correlation, means, and standard deviations for ER and maladaptive symptoms for both samples.

Variables	M (SD)	1	2	3	4	5	6
1. INT	7.52 (3.26)	1					
2. EXT	5.65 (3.83)	0.06	1				
3. NSSI	2.10 (2.09)	0.16 **	0.54 **	1			
4. DER	2.77 (0.75)	0.29 **	0.43 **	0.38 **	1		
5. SER	3.21 (0.84)	0.32 **	−0.37 **	0.18 **	−0.07	1	
6. IER	2.98 (0.79)	−0.09	0.11	0.001	0.14 *	−0.19 **	1

Note. INT = internalizing problems; EXT = externalizing problems. ** Correlation is significant at the 0.01 level. * Correlation is significant at the 0.05 level.

**Table 2 healthcare-12-00341-t002:** Means, standard deviations, and bivariate correlations between the variables for Iranian (n = 117) and Belgian (n = 107) samples.

Variables	M (SD) Belgium	M (SD) Iran	1	2	3	4	5	6
1. INT	6.39 (3.55)	8.55 (2.60)	1	0.32 **	0.27 **	0.31 **	0.21 *	−0.03
2. EXT	7.57 (3.53)	3.81 (3.31)	0.22 *	1	0.57 **	0.48 **	−0.26 **	0.11
3. NSSI			0.39 **	0.33 **	1	0.42 **	−0.13	−0.03
4. DER	2.86 (0.76)	2.68 (0.73)	0.39 **	0.38 **	0.32 **	1	0.04	0.13
5. SER	2.72 (0.71)	3.65 (0.70)	0.15	−0.03	0.19 *	−0.04	1	0.05
6. IER	3.19 (0.69)	2.79 (0.82)	0.02	−0.14	−0.2 *	0.09	−0.19 *	1

Note. INT = internalizing problems; EXT = externalizing problems; Upper half correlations = Iran; Lower half correlations = Belgium. ** Correlation is significant at the 0.01 level. * Correlation is significant at the 0.05 level.

**Table 3 healthcare-12-00341-t003:** Mean scores for ER Modes and internalizing and externalizing problems by country.

Variables	Iran (n = 117)	Belgium (n = 107)	F
	M (SD)	M (SD)	
DER	2.68 (0.74)	2.86 (0.76)	3.215
SER	3.65 (0.71)	2.72 (0.71)	95.9 **
IER	2.79 (0.82)	3.19 (0.69)	15.04 **
Internalizing	8.55 (2.6)	6.39 (3.55)	27.03 **
Externalizing	3.81 (3.3)	7.57 (3.53)	67.5 **

** Difference is significant at the 0.01 level.

**Table 4 healthcare-12-00341-t004:** Hierarchical regression analysis for the role of age, gender, country, and ER modes in the prediction of NSSI and internalizing and externalizing problems.

		**NSSI**			
		*β*	SE	B	t
Model 1	Age	−0.07	0.13	−0.13	−0.99
	Gender	−0.02	0.26	−0.09	−0.36
	Country	−0.44	0.32	−1.85	−5.76 **
Model 2	Age	−0.04	0.12	−0.08	−0.62
	Gender	−0.02	0.24	−1.13	0.46
	Country	−0.46	0.33	−1.92	−5.69 **
	DER	0.35	0.16	0.97	5.96 **
	SER	0.05	0.16	0.13	0.81
	IER	−0.15	0.16	−0.41	−2.59
		**Internalizing problems**			
		*β*	SE	B	t
Model 1	Age	0.13	0.21	0.05	0.62
	Gender	−1.89	0.4	−0.29	−4.76 **
	Country	2.53	0.49	0.4	5.19 **
Model 2	Age	0.14	0.2	0.05	0.74
	Gender	−1.56	0.37	−0.24	−4.19 **
	Country	2.01	0.52	0.32	3.91 **
	DER	1.37	0.25	0.32	5.47 **
	SER	0.68	0.26	0.18	2.65 **
	IER	−0.23	0.24	−0.06	−0.94
		**Externalizing problems**			
		*β*	SE	B	t
Model 1	Age	−0.47	0.24	−0.14	−1.93
	Gender	0.42	0.46	0.05	0.9
	Country	−4.45	0.57	−0.57	−7.82 **
Model 2	Age	−0.44	0.22	−0.13	−1.97 *
	Gender	0.88	0.42	0.11	2.08 *
	Country	−3.58	0.58	−0.46	−6.14 **
	DER	2.04	0.28	0.39	7.22 **
	SER	−0.66	0.29	−0.14	−2.26 *
	IER	−0.13	0.27	−0.03	−0.47

Note. NSSI Model: F = 14.56 **, R^2^ = 0.16, Adjusted R^2^ = 0.15 (Model 1), F = 15.30 **, R^2^= 0.30, Adjusted R^2^ = 0.28 (Model 2). Internalizing and Externalizing Model: F = 3.44, R^2^ = 0.04, Adjusted R^2^ = 0.03 (Model 1), F = 13.41 **, R^2^ = 0.27, Adjusted R^2^ = 0.27 (Model 2). NSSI represented as a binary variable, where a value of 1 signifies the presence of self-injury, while a value of 0 indicates the absence of self-injury. * Difference is significant at the 0.05 level.

## Data Availability

Data are contained within the article.

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
