# Peer review of "A Comparative Analysis of Emotional Regulation and Maladaptive Symptoms in Adolescents: Insights from Iran and Belgium"

_healthcare, 2024, doi:10.3390/healthcare12030341_

Round 1

Reviewer 1 Report

Comments and Suggestions for Authors

Dear authors,

I would like to appreciate your work. I usually refer to the introduction line by line as it is the most convenient way to improve this section. Later in other sections, I would like to give a general opinion and if needed, specifically point out the section I want to be clarified. Reading through your introduction, I found it clear and to the point except for one issue:

In the hypothesis section, you have again referred to some other studies to state your current hypothesis. Although citing previous research is good, I believe the hypotheses should be more clear and concise. I suggest moving those references from your hypotheses to the introduction section rather than mentioning them there. My rationale is that as a reader, I tend to find hypotheses short enough yet informative. You can cover the references in the introduction.

  1. Method:

The critical question for me in reviewing your work is this: Did you conduct any power analysis before the study? Stating whether you performed a power analysis is vital in this context. Based on your variables and tests, it appears you have adequate power to run your analyses. However, reporting a power analysis is a key element to include. Consider this for any further studies you conduct.

You have two different groups with age discrepancies. Since Belgium encompasses substantial diversity in areas like religion, economic status, and school systems, how did you ensure these variations would not impact your results? I wonder specifically about the potential influence of economic differences given that research has shown emotion regulation is affected by socioeconomic status (e.g., Heilman, 2016; Troy, 2017). Those from lower socioeconomic strata tend to benefit more from emotion regulation in terms of mental health.

I did not understand how you obtained your 30.83% Belgian participant sample. You should explain this more clearly.

2.2 Measures:

The SDQ questionnaire explanation is overly lengthy. I suggest shortening this description, although it is not imperative. When targeting scientific audiences, such elaborations are unnecessary.

Why did you not include SDQ subscales and only utilize total scores? Please provide the rationale.

For the ERI questionnaire, based on your text, there is no validated Persian version. If so, do you believe your sample size can validate your translation adequately for the current study?

Since you gave details about the SHI for both languages, you may want to provide more specifics regarding validation for the Iranian and Belgian versions.

Statistical Analysis:

As you chose MANOVA, did you assess the normality of your obtained data distributions? Considering MANOVA's robustness to minor normality deviations, such violations may alter your Type I and Type II error interpretations. If yes, provide details on the normality confirmation procedures.

3 Results:

Clarify whether Table 1 displays cumulative data for both groups or separated data.

In the PDF version I reviewed, Table 2 is confusing with Iran and Belgium labels but has no corresponding data. Double-check this table is presented accurately.

Use consistent backgrounds and enhance the quality of Figures 1 and 2.

Discussion:

You have again briefly restated the results. However, given the extent of your cross-cultural differences, please expand further on explaining what may account for this variation. For instance, when aligning your data with Mesquita and Dadkhah's findings, what could explain similarities or discrepancies to this degree? In your opinion, what specifically drives such patterns? Adding more elucidation here would be beneficial.

As you compare two countries, provide more empirical evidence for statements linking your observations to cultural values (e.g., L323-324 “The observed higher prevalence of IER among Belgians reflects the cultural emphasis on self-expression, self-reliance, and the pursuit of individual goals”).

Reviewer 2 Report

Comments and Suggestions for Authors

The authors investigate a topical problem for both fundamental and applied psychology (emotion regulation in the context of personality adaptive potential) and identify significant aspects in its study. The article makes an attempt at a cross-cultural analysis of the relationship between the phenomena under consideration, which is certainly a merit of the work. A study has been carried out using a variety of mathematical statistics and mathematical modeling methods.

The content of the article is significant in terms of developing practical recommendations for the prevention and optimisation of adolescents' psychological well-being.

However, the text of the article must highlight the limitations of the study due to organisational or methodological issues (for example, the extent to which the limitation of the sample of young people to 100 people in each comparison group enables us to assess the presence of cultural specificities in this sample). How was the typicality of the carriers of the culture determined? How reasonable is it to compare samples that are not asymmetric in terms of age boundaries (13–15 and 14–19 years old), what features related to this could affect the results obtained? etc.).

Reviewer 3 Report

Comments and Suggestions for Authors

Line 97: Perhaps here under the heading of ‘Present study’ you could explicitly list and number research hypotheses and questions? For example: “Research Hypothesis 1: There will be a significant negative relationship between IER and psychological symptoms.”

Line 115: I believe “ER [or a review” should be “ER [for a review.”

Line 144: It seems redundant to have both ‘self-reported’ and ‘self-report.’

Line 146: You mention “construct validity (r = 0.70 for the total score).” R is not a mention of construct validity.

Line 159. What does ‘back-translated’ mean?

The heading for Table 3 can be improved. Something like: ‘Mean scores for ER modes and internalizing and externalizing problems by country’

Line 323: Should ‘Hofstede insights country comparison tool’ be capitalised?

Line 338: ‘underscores’ should be singular (underscore)

Line 380: “The observation that NSSI was correlated with heightened.” Remove the word ‘was.’

Please mention strengths and weaknesses in the discussion.

Reviewer 4 Report

Comments and Suggestions for Authors

Thank you for the opportunity to review the manuscript, “A Comparative Analysis of Emotion Regulation and Maladaptive Symptoms in Adolescents: Insights from Iran and Belgium." This study examined the relationship between emotion regulation and maladaptive psychological symptoms in two countries. The main strength of the study was the examination of the relationship in two culturally different settings, giving a deeper understanding of the differences and similarities of the link between emotion regulation and the maladaptive functioning of adolescents. The used literature is up-to-date, and the manuscript’s results are reproducible based on the details given in the methods section. The main limitation is the relatively small number of participants per country, which limits the generalization of the findings. But the manuscript is clearly structured and well-written. Thus, with certain revisions that I stated here-specific comments that may help the authors with the revision of the manuscript-I think it has the potential for publication.

Introduction

  1. In the introduction, lines 82-96, it would be good to add a description of Belgian families and culture the same as that given for Iranian families.
  2. In the introduction, there is a paragraph about the present study (lines 98-118). “Moreover, given 105 previous research [e.g., 42] indicating a stronger correlation between expressive suppression and negative mental health indicators in Western cultural groups, we expect potential cross-cultural differences in the correlation between SER and psychological symptoms.” Please specify what cross-cultural differences you expected. Also, please mark the hypotheses clearly with a number or letter. Also, please refer to your hypotheses in the discussion section. Are hypotheses confirmed or not, and why?
  3. Line 114-115 “In light of previous research describing age-related differences in ER [or a review, see 48]..." Please check the brackets for spelling. Maybe is for instead of or.

Materials and Methods

  1. Line 132: Why do you mention religious orientation if this variable was not used in the analysis? Maybe it would be good to add the frequency of answers for this variable to better describe the sample.
  2. For all measures, please add Cronbach alpha for all used subscales or scales by country. In addition, describe how the total score for all used measures was calculated and the meaning of lower or higher results for the formulated scores.
  3. Please add the response rate for adolescents and parents’ participation. For all the contacted parents, how many of them agreed to participate in the study? Please add this information to the procedure section.

Results

  1. Please add data about M and SD to Table 2.
  2. Related to Table 3. Why was the difference in NSSI between two countries not tested? Please test the difference and add the results to the table.
  3. Related to Table 4. How was the internalizing and externalizing problems score used in the regression analysis calculated? Why were the separate scores for both problems not used in the regression analysis?
  4. Regarding the mediation analysis, please provide a description of the first tested model. I suppose that both direct and indirect relationships are tested, and then insignificant ones are removed, but this needs to be clearly stated in the manuscript.
  5. Please revise and match the labels of variables on Figure 1. and Figure 2. with the abbreviations used (e.g., ER, NSSI, DER).
  6. Can you check this out? “The values of the standardized regression weights for direct effects were statistically significant for all paths in the model, except for the path from SER to NSSI (β = 0.18, p > 0.05).” (lines 293-295), since this does not correspond to the results in Figure 2.

Discussion

  1. Lines 304-307 “The primary objective of this study was to illuminate cultural differences in the relationship between distinct ER modes and adolescent psychopathology based on the SDT framework and to examine the mediating role of internalizing and externalizing problems in the association between ER modes and NSSI in two different cultural groups.” Please replace “illuminate” with a different word or verb (e.g., examine, explore).
  2. In the discussion, please refer to the hypotheses and state whether they are confirmed.
  3. We need more explanation on why certain cultural differences emerged based on the theory you grounded your manuscript in and other relevant sources.
  4. Please add and describe the study's limitations and suggestions for future research.
  5. Please formulate the conclusion, taking into account the size of the sample and study's limitations.

At the end, I congratulate the authors for their effort to conduct the study and wish them all the best.

Comments on the Quality of English Language

Minor editing of English language required. 

Round 2

Reviewer 1 Report

Comments and Suggestions for Authors

Dear authors,

I express my gratitude for the prompt response with respect to the issues that I raised about your manuscript a while ago. Your positive response, along with the sincerity expressed in taking feedback, is indeed appreciated.

Upon further recollection, I suppose you adeptly handled those issues at hand, particularly when it came to the questionnaire parts. While I argue that the inclusion of a subscale, as highlighted in literature by Kroc (2021), Reise (2013), Hinz (2005), and Bech (2003), requires some reflective thought, to what extent such elements were previously dealt with through current revision is acknowledged and find it satisfactory for this review.

I would, however, venture to draw your attention to the structural layout of the manuscript. Certain sections seem excessively long, which may be due to the peculiarities of the PDF file format. For example

, in line 163 after Sousa, the sentence could be broken into a new line to appear more readable and make more sense.

In the row, the statistics part from lines 208 through 232 would be effective if at least two paragraphs were used. The same suggestion of structural improvement, therefore, applies herein to the segments running from line 390 through 423 and between 441 and 465.

Here, it is essential to mention that the above briefing does not necessarily mean a reduction in content but optimum structuring of the content so that it can be more reader-friendly and interesting.

I would also recommend one last edit or proofread in order to make finer tweaks in grammar. For example, a sentence of the kind "These results are consistent…" might need to be reviewed for better language and clarity. Beyond any doubt, all this relentless attention to detail will surely take your academic work to another level of quality and outreach.

In conclusion, I appreciate your continuous commitment to academic quality and eagerly await the development of your manuscript.

Comments on the Quality of English Language

some minor changes and editing is required

Reviewer 4 Report

Comments and Suggestions for Authors

The revised version of the paper is very good, and almost all my remarks have been considered. I only have a few additional comments that I think will help to improve the paper.

2 Remark: In Hypotheses 2 and 3, please specify the direction of expected mediation and the effects of age, gender, and country. Please try to avoid using the term "influence.”.

9 remark: Table 4. Add notation about how NSSI was coded since it is a binary variable. It is going to help future readers with the interpretation of the table.

17 remark: “Our study contributes to the growing body of research on ER modes and their impact on adolescent psychopathology from an SDT perspective”  (line 466-467) The word “impact” in this sentence should be replaced with another word (e.g., association, relationship, link). And this should be put into context in relation to the sample characteristics and sampling method.

Small details:

Line 388, “findings of Benita," please add a reference.

Line 460: Can you be more specific about what you mean under “EMA technologies"?

Line 456: in my version, "5. Conclusion" is glued to the last sentence of the discussion section. Maybe you can check that out.

All the best!

Comments on the Quality of English Language

You can check the text once all corrections are done.
